# Relationships between Bone Turnover Markers and Factors Associated with Metabolic Syndrome in Prepubertal Girls and Boys

**DOI:** 10.3390/nu14061205

**Published:** 2022-03-12

**Authors:** Wojciech J. Bilinski, Anna Stefanska, Lukasz Szternel, Katarzyna Bergmann, Joanna Siodmiak, Magdalena Krintus, Przemyslaw T. Paradowski, Grazyna Sypniewska

**Affiliations:** 1Department of Orthopaedics, KoMed, Poddebice Health Center, 85067 Poddebice, Poland; bilinski_wojtek@wp.pl; 2Department of Laboratory Medicine, Collegium Medicum, Bydgoszcz, Nicolaus Copernicus University, 87110 Torun, Poland; l.szternel@cm.umk.pl (L.S.); bergmann@cm.umk.pl (K.B.); joanna.pollak@cm.umk.pl (J.S.); krintus@cm.umk.pl (M.K.); odes@cm.umk.pl (G.S.); 3Department of Surgical and Perioperative Sciences, Division of Orthopedics, Sunderby Research Unit, Umea University, Sunderby Central Hospital of Norrbotten, SE-971 80 Lulea, Sweden; przemyslaw.paradowski@umu.se; 4Clinical Epidemiology Unit, Orthopedics, Department of Clinical Sciences Lund, Lund University, SE-221 85 Lund, Sweden

**Keywords:** bone turnover markers, metabolic syndrome, insulin resistance, C-reactive protein

## Abstract

The associations between individual components of metabolic syndrome (MetS) and bone health in children are complex, and data on this topic are sparse and inconsistent. We assessed the relationship between bone turnover markers and markers of the processes underlying MetS (insulin resistance and inflammation) in a group of presumably healthy children aged 9–11 years: 89 (51 girls, 38 boys) presenting without any features of MetS and 26 (10 girls, 16 boys) with central obesity and two features of MetS. Concentrations of glucose, triglycerides (TG), HDL cholesterol (HDL-C), C-reactive protein (CRP), HbA1c, total 25-hydroxyvitamin D (25(OH)D), intact-P1NP (N-terminal propeptide of type I procollagen), CTX-1 (C-terminal telopeptide of type I collagen) were assayed and insulin resistance was assessed (HOMA-IR). BMI centile, waist circumference (WC) and blood pressure were measured. The presence of MetS in girls resulted in significantly lower concentrations of CTX-1 and a trend to lower CTX-1 in boys. The concentrations of bone formation marker i-P1NP were not affected. Among the features associated with MetS, HOMA-IR appeared as the best positive predictor of MetS in girls, whereas CRP was the best positive predictor in boys. A significant influence of HOMA-IR on the decrease in CTX-1 in girls was independent of BMI centile and WC, and the OR of having CTX-1 below the median was 2.8-fold higher/1SD increased in HOMA-IR (*p* = 0.003). A weak relationship between CTX-1 and CRP was demonstrated in girls (r = −0.233; *p* = 0.070). Although TG, as a MetS component, was the best significant predictor of MetS in both sexes, there were no correlations between bone markers and TG. We suggest that dyslipidemia is not associated with the levels of bone markers in prepubertal children whereas CRP is weakly related to bone resorption in girls. In prepubertal girls, insulin resistance exerts a dominant negative impact on bone resorption, independent of BMI centile and waist circumference.

## 1. Introduction

Childhood obesity and concomitant metabolic abnormalities may impair bone development leading to decreased bone mineral content, bone mineral density (BMD) and bone strength [1,2]. Metabolic syndrome (MetS), a cluster of components associated with increased cardiovascular risk (increased waist circumference, fasting hyperglycemia, increased triglycerides and blood pressure, decreased high-density lipoprotein cholesterol) has been shown to adversely affect bone mass and bone mineral density in adolescents [3,4].

An earlier study performed in overweight adolescents (10–16 years old) with MetS indicated that BMD/kg body weight was significantly reduced at different skeletal sites [4]. Nevertheless, data on the effects of individual MetS components on bone health in adolescents are sparse and controversial.

The relationship between bone mineral content and insulin resistance (HOMA-IR) was assessed in a population-based sample of Korean adolescents aged 10–19 years [5]. It was shown that higher HOMA-IR was associated with a decrease in BMC (bone mineral content) in male adolescents aged 13–19 years but the associations were not significant in younger female and male adolescents (10–12 years) [5]. By contrast, in another study, insulin resistance (HOMA-IR) in 9–12 years old girls was shown to be the main predictor of regional bone outcomes, in particular, trabecular bone [6].

In one recent study, increased levels of TG and hypertension in young girls were not associated with any total body or regional bone outcomes [6]. In the other however, it was indicated that adolescent girls (10–16 years) with increased blood pressure and hypertriglyceridemia showed reduced BMD in several skeletal sites, whereas in males, increased triglycerides had no effect on BMD [4,7].

The individual components of MetS may occur together, driven by mechanisms leading to insulin resistance and linked to inflammatory processes. The prevalence of MetS is greater in male than female adolescents and varies with age. In addition, MetS is linked to obesity-related diseases, including non-alcoholic fatty liver disease and kidney function [8]. The effect of MetS occurrence on bone health in adolescents has been a subject of interest since optimal bone accrual in this period is critical to reach an optimal peak bone mass. Previous data not only are incoherent but also did not report on the effects of individual MetS components on bone metabolism. There is only one report to date on the relationship between components of MetS and bone turnover biomarkers in adolescents [7].

Circulating bone turnover markers reflect the metabolism of the entire skeleton. Biochemical markers are components of the bone matrix released into the circulation during bone formation or bone resorption. Out of several bone turnover markers, only P1NP (N-terminal propeptide of type I procollagen, a bone-formation marker) and CTX (C-terminal telopeptide of type I collagen, a bone-resorption marker) were recommended as reference biomarkers to use when investigating bone metabolism in adults [9]. P1NP reflects the synthesis of new bone collagen, and its serum level displays greater dynamic changes than other bone formation markers in response to changes in bone metabolism. CTX is released from the bone matrix during resorption by osteoclasts. P1NP is characterized by little biological variation and can be measured in a random blood sample, but CTX displays higher diurnal and biological variation [10].

Both P1NP and CTX are currently used for the diagnosis and management of skeletal diseases in children, although due to rapid skeletal growth and the physiological changes associated with puberty, appropriate reference intervals should be applied for the interpretation of results [9,10,11,12]. Important here is that the reference intervals for some bone turnover markers, including P1NP and CTX, have been established in representative groups of different populations of children and adolescents [12,13,14,15,16,17].

No study to date has examined the effect of factors associated with MetS on bone turnover in prepubertal children. Accordingly, we aimed to assess the relationships between the reference markers of bone turnover (i-P1NP and CTX-1) and markers of the processes underlying MetS, including insulin resistance and low-grade inflammation in a group of presumably healthy prepubertal school children aged 9–11 years.

## 2. Materials and Methods

### 2.1. Characteristics of Study Participants

From a cohort of 284 presumably healthy school children aged 9–11, enrolled in a previous cross-sectional study [18], a subset of 115 children was selected for whom complete data on biochemical parameters, markers of bone metabolism, blood pressure and anthropometric parameters were available. In the previous study, the children were selected on the basis of age (9–11 years old) from four primary schools in the one suburban region. The children’s age was the first inclusion criterion taken into consideration in accordance with AAP guidelines, which recommend performing the first lipid profile screening in children who are not in a high-risk group [19]. The second inclusion criterion was a fasting state (a minimum of 8 h since last meal) before blood drawing. Whilst school nurses and internal medicine specialists participated in the recruitment process, the general health of the child on the day of blood draw was subjectively evaluated by their parents. Immediately following blood collection, the blood samples were transported to the laboratory and centrifuged to obtain serum. Whole-blood samples were collected for HbA1c evaluation.

In the subset of 115 children included in this study, 89 subjects (51 girls and 38 boys) presented without any features of metabolic syndrome (MetS) and 26 subjects (10 girls and 16 boys) presented with central obesity and at least two other features of MetS (fasting hyperglycemia, fasting dyslipidemia and/or increased blood pressure). Tanner staging (TS) was also recorded based on self-assessment reports. Among girls without MetS, 48 (94%) reported TS1 and 3 (6%) TS2. In the group with MetS, TS2 was reported by 3 girls (30%). Among boys from the group without MetS, 33 reported TS1, 3 (8%) TS2 and 2 (5%) TS3, while in the group with MetS, 5 boys (31%) reported TS2.

The compulsory physical activity of study participants at school was a minimum of 3 h per week, as outlined in state physical education directives for curricula in public schools. According to Statistical Office data, the average child aged 2–14 spends about 2.4 h in front of a TV screen and/or computer monitor. Physical activity is reported by 85% of children aged 6–14 [18].

Children did not receive any supplements or medicines which might have affected their bone metabolism or vitamin D levels.

### 2.2. Laboratory and Anthropometric Measurements

Venous blood samples were taken in a fasting state in order to obtain serum, and whole blood was collected for glycated hemoglobin (HbA1c) assay. The following parameters: glucose, triglycerides (TG), HDL-cholesterol (HDL-C), C-reactive protein (CRP) and HbA1c, were measured immediately, as previously described [20]. Total 25-hydroxyvitamin D (25(OH)D), intact-P1NP, CTX-1 and insulin were assayed in previously deep-frozen serum samples, as reported earlier [20]. Total 25(OH)D, i-P1NP and CTX-1 were analyzed on an IDS-iSYS automated platform (Immunodiagnostic Systems Holding PLC, Boldon, UK). Insulin resistance was assessed on the basis of the calculated homeostasis model assessment (HOMA-IR) and bone turnover index (BTI) was calculated as previously described [20]. All laboratory tests were performed at the Department of Laboratory Medicine, Collegium Medicum, Nicolaus Copernicus University in Torun. Anthropometric measurements were performed on the same day as blood samples were taken and blood pressure was measured by a qualified nursing team after 5 min of rest in a sitting position. BMI (body mass index) and BMI percentiles were calculated with the use of an online calculator for children and adolescents in Poland, as reported previously [18].

### 2.3. Decision Criteria for Study Participants

BMI percentile classifications were performed in accordance with the guidelines of the International Obesity Task Force: underweight BMI < 5; overweight ≥ 85 and <95; obesity ≥ 95 percentile [18]. Participants were divided according to sex into two groups: without MetS—presenting without any component of metabolic syndrome, and MetS group—presenting with central obesity plus any other two components of MetS. The components of MetS were defined according to the National Cholesterol Education Program Adult Treatment Panel III definition modified for age [8]. Central obesity was defined by a waist circumference ≥ 90th percentile of WC by sex and age for European population, fasting hyperglycemia as blood glucose ≥ 100 mg/dL, increased blood pressure as systolic blood pressure (SBP) or diastolic blood pressure (DBP) ≥ 90th percentile of BP by sex, height and age [21]. Fasting dyslipidemia was defined as TG ≥ 110 mg/dL and low HDL-C as <40 mg/dL [8].

Insulin resistance was recognized on the basis of the calculated homeostasis model assessment, HOMA-IR, at a cut-off value of ≥3.0, in accordance with the reference ranges suggested by Shashaj et al. for European children [22]. Fasting insulin ≥ 12 µU/mL was accepted as hyperinsulinemia and CRP concentration > 3 mg/L as increased. BTI was calculated based on the results of i-P1NP and CTX-1 concentrations as previously described [20]. Positive BTI indicated that bone formation prevails over bone resorption, while a negative BTI value reflected the predominance of bone resorption [11]. Concentrations of 25(OH)D < 20 ng/mL were accepted as a deficiency according to currently applied recommendations [23].

The study has been carried out in accordance with the Declaration of Helsinki and was approved by the Collegium Medicum Bioethics Committee at the Nicolaus Copernicus University (KB 338/2015, annex 487/2019). Parental written consent forms were obtained for all participants before inclusion in the study.

### 2.4. Statistical Analysis

The data were presented as medians and 25th and 75th percentiles (non-Gaussian distribution). The Shapiro–Wilk test was applied to test the normality of the variables before and after natural log transformation. The variables were compared using the Mann–Whitney *U* test. Binary data in 2 × 2 tables were evaluated by Fisher’s exact test. Spearman rank correlations were evaluated. Parameters with non-Gaussian distribution were normalized by natural log transformation for multiple linear regression and univariate logistic analysis. In multiple linear regression analysis, BMI centile, WC and HOMA-IR values were assayed as independent variables and CTX-1 as a dependent variable. In all the logistic models, odds ratios (ORs) were calculated for a 1 standard deviation (SD) increase in independent variables. The significance of coefficients in the logistic models was tested using Wald chi-squared statistics. The goodness of fit of the models was evaluated using the Hosmer and Lemeshow chi-squared test. To correct *p* values, the Benjamini–Hochberg procedure was applied to risk factor results to reduce the potential for type 1 error. The level of statistical significance was set as 0.05 (Statistica 13.3, StatSoft or MedCalc statistical software).

## 3. Results

Subjects were compared according to sex. Clinical and biochemical characteristics of study groups are displayed in Table 1. In children with MetS, irrespective of sex, 30% presented with the same degree of maturation (TS2). Boys with MetS were significantly higher (*p* = 0.007), which was not observed among girls. Overall, in girls and boys the presence of MetS did not affect the levels of glucose, HbA1c and 25(OH)D. Occurrence of MetS had a significant impact on bone turnover. The concentrations of bone resorption marker CTX-1 were significantly lower in girls with MetS and showed a trend to lower values in boys with MetS. On the contrary, the presence of MetS did not affect either i-P1NP—the bone formation marker concentrations. Calculated bone turnover index values in subjects with MetS reflected the predominance of bone formation over resorption, particularly in girls. Lower CTX-1 levels in children with MetS were accompanied by significantly higher insulin, HOMA-IR and higher CRP values.

The occurrence of features of metabolic syndrome in girls and boys was presented in Figure 1. Among the components of MetS in both girls and boys, fasting hypertriglyceridemia was the most prevalent, followed by increased systolic blood pressure, apart from central obesity. Fasting hyperglycemia was observed less frequently and mostly in girls, whereas decreased level of HDL-C was found only in 20% of girls and very rarely in boys (6%).

Fasting hyperinsulinemia and an increased value of HOMA-IR were the most often observed features associated with MetS, irrespective of sex (Table 2). Increased CRP and decreased 25(OH)D levels were observed less frequently.

Logistic regression analysis was performed for the assessment of the relationship between individual risk factors, biomarkers of bone metabolism (CTX-1, i-P1NP) and the presence of MetS (Table 3). This analysis revealed sex-specific differences in terms of the predictors of MetS. The models in the univariable analysis indicated that values of HOMA-IR and levels of TG in girls were the best positive predictors of having MetS. A significant negative association between the presence of MetS and CTX-1 was found. On the contrary, in boys, TG and CRP levels appeared as the best significant positive predictors of having MetS. No significant associations between P1NP and the presence of MetS were observed in girls or boys.

The analyses of Spearman correlations were performed between metabolic factors and CTX-1, as the bone turnover biomarker associated with the presence of MetS. An inverse correlation between CTX-1 and BMI centile (r = −0.33; *p* = 0.009), CTX-1 and HOMA-IR (r = −0.34; *p* = 0.008) and CTX-1 and insulin concentration (r = −0.33; *p* = 0.01) was found in the whole group of girls. The relationship between HOMA-IR and CTX-1 in girls was also assessed after adjustment for BMI centile and WC in multiple linear regression analysis (Table 4). This analysis revealed that the influence of HOMA-IR on the decrease in CTX-1 remained significant, being independent of BMI centile and WC.

In-depth analysis demonstrated the impact of HOMA-IR, BMI centile and WC on the CTX-1 values below the median (<1.50 ng/mL) calculated for the whole study group (*n* = 115). We found that in girls, OR of having CTX-1 < 1.50 ng/mL was almost threefold higher per 1 SD change in HOMA-IR, whereas this was not the case for BMI centile and WC. Based on the result of Nagelkerke R2, the model with HOMA-IR explained 22% of the variation in the occurrence of CTX-1 < 1.50 ng/mL in girls. Similar relationships were not observed in the group of boys (Table 5).

Regarding biomarkers of inflammation, CRP was significantly higher in children with MetS, particularly in boys, however we did not find significant correlations between bone turnover markers and CRP. In the present study, a very weak inverse relationship between CTX-1 and CRP was demonstrated only in girls (r = −0.233; *p* = 0.070), which became significant and stronger in those with CRP > 0.5 mg/L (r = −0.46; *p* = 0.009; *n* = 31). We have not found any correlation between CTX-1 and CRP in boys.

## 4. Discussion

Metabolic abnormalities in children with overweight and obesity may influence bone mineral content and bone structure, therefore assessment of bone health in childhood in order to prevent the inadequate bone mass accrual is receiving growing attention [24].

Studies in adult populations on the relationship between MetS and bone health have reported inconsistent results, demonstrating a positive or negative relationship with bone mineral density [25,26]. These studies clearly indicated that the association between MetS and bone health cannot be considered without taking into account sex differences, adjustment for BMI or waist circumference and correlations between individual components of MetS and bone mineral density.

In an earlier cross-sectional study performed on Brazilian adolescents aged 10–16 years, the impact of MetS on bone mineral density and bone turnover markers was examined [7]. This study indicated that the presence of MetS was not associated with decreased lumbar spine or femoral neck bone mineral density, but only with significantly decreased BMD/kg of body weight.

In the above-mentioned research, the effect of MetS on the levels of bone turnover markers in adolescents was assessed [7]. It was found that in subjects with MetS, the concentrations of bone-formation (osteocalcin (OC), bone alkaline phosphatase (b-ALP)) and bone-resorption markers (CTx), except osteocalcin in boys, were significantly decreased.

The present study included prepubertal girls and boys with and without MetS in whom we assessed two bone turnover markers, i-P1NP and CTX-1, and their relationships with MetS and factors associated with MetS (CRP, HOMA-IR, fasting hyperinsulinemia). We demonstrated that the presence of MetS in girls resulted in significantly lower concentrations of CTX-1 and only a trend to lower CTX-1 levels in boys. By contrast, the concentrations of bone formation marker i-P1NP in girls and boys with MetS were not affected. In children with MetS, the BTI value indicated the predominance of bone formation over resorption, particularly in girls.

The incompatibility of results between two studies may be due to different bone formation markers assessed. I-P1NP, which we measured, is characterized by low biological variation (8.4–9.3%) [27] and reflects the synthesis of new bone collagen, whereas osteocalcin and b-ALP determined in the above-referenced study reflect mineralization of the bone matrix. The other explanation for these discrepancies may be due to the fact that children included in our study were younger, and at this age (9–11 years), the levels of P1NP are very stable (present a very narrow range of changes). On the other hand, OC, as has been shown previously by others, starts to increase after 11 years of age in girls and after 13 years of age in boys, and its concentrations change within a wide range until 14 years and 16 years in females and males, respectively [12].

We have shown that in girls, CTX-1 level was significantly and negatively related with the presence of MetS. Moreover, we demonstrated sex differences in relation to the individual predictors of MetS.

In our study, HOMA-IR appeared as the best positive predictor of MetS in girls, whereas CRP level was the best positive predictor of MetS in boys. This contradicts with the observations of others, which have shown that CRP was the best significant predictor of MetS in adolescent girls but not in boys [28]. The referred to study included Spanish adolescents aged 12–16 years, and only 11 males and 4 females with MetS.

Noteworthy, is that HOMA-IR has been shown previously as the main predictor of regional bone outcomes in young girls [6]. We have found that fasting hyperinsulinemia and increased values of HOMA-IR were the most often observed features associated with MetS in both sexes, however, more detailed analysis revealed the significant impact of HOMA-IR on bone resorption solely in girls.

Multiple linear regression analysis performed in the whole group of girls revealed that the significant influence of HOMA-IR on the decrease in CTX-1 was independent of BMI centile and waist circumference. Similar relationships were not observed in boys. Furthermore, only in girls we were able to demonstrate the significant impact of HOMA-IR on the CTX-1 levels below the median (<1.50 ng/mL). The OR of having CTX-1 below the median was almost 3-fold higher per 1 SD increase in HOMA-IR (*p* = 0.003) and the model explained 22% of the variation for the occurrence of CTX-1 < 1.50 ng/mL.

Although CRP was significantly higher in children with MetS and was the best predictor of MetS in boys, we did not find correlations between bone turnover markers and CRP. In our previous work, a trend towards an inverse correlation between CRP and CTX-1 was observed (r = −0.131; *p* = 0.083) in a group of 182 presumably healthy children aged 9–11 years (unpublished results). In the present study, a weak inverse relationship between CTX-1 and CRP was demonstrated only in girls (r = −0.233; *p* = 0.070), which became significant and stronger in those with CRP > 0.5 mg/L (r = −0.46; *p* = 0.009). The association between CRP and CTX-1 may be partially explained by BMI, because we found significant positive correlations between CRP levels and BMI centile among girls and boys (r = 0.421; *p* < 0.001 and r = 0.495; *p* < 0.001, respectively).

The effect of dyslipidemia on bone health was evaluated in the observational studies, which suggested that dyslipidemia in adults, and most of all, high total cholesterol, and to a smaller extent, increased triglyceride concentrations, is associated with low bone mass and increased risk of fractures [29,30,31].

Interestingly, in the present study, TG level was the best significant predictor of MetS in both sexes. We did not find, however, any correlations between bone turnover markers and TG or HDL-C despite the significantly higher TG and significantly lower HDL-C in children with MetS. We assume therefore, that the decrease in CTX-1 in children with MetS may result from factors linked with dyslipidemia such as hormonal status and systemic inflammation, as suggested by others [24].

The effect of increased blood pressure on bone metabolism is a matter of controversy, and in adult populations it was associated positively with total hip BMD but not with spine BMD [32], which contradicts the findings of a meta-analysis showing that blood pressure is negatively associated with BMD and positively with increased risk of osteoporosis [33]. It is suggested that high blood pressure increases calcium excretion, elevates parathyroid hormone level and thus increases bone resorption [34].

The data on the influence of hypertension on bone health in children refer mainly to young girls and are conflicting [6,7]. In the previous study conducted in adolescents, negative correlations between SBP and OC and b-ALP but not with CTx have been observed in females but not in males with MetS [7]. The small number of children with MetS included in our research makes it impossible to link blood pressure directly with bone turnover. We concur with the hypothesis that hyperinsulinemia, which may lead to excessive sodium reabsorption, potentially contributes to elevated blood pressure [35]. This is in line with our findings indicating blood pressure as a predictor of having MetS, particularly in girls. Hyperinsulinemia and degree of insulin resistance, which underlie the occurrence of MetS, have the most influence on the level of bone resorption markers in prepubertal girls, independent of BMI centile and waist circumference.

The results of our study should be interpreted in the light of some limitations. We were unable to collect detailed dietary reports, however, the study excluded underweight children and those taking vitamins and medicines that might affect their bone metabolism. Only the information on compulsory physical activity at school was available. As another limitation of the present study, we have to mention the small sample size, in particular the very small numbers of girls and boys with metabolic syndrome, which makes the problem of rejecting the null hypothesis in the comparison of subgroups and drawing general conclusions quite difficult.

On the other hand, the concentrations of assessed bone markers are quite stable in this narrow age range (9–11 years), thereby demonstrating that observed differences may be, only to some small extent, due to physiologic fluctuations during growth. Bone accrual in children with excess weight is affected by several different humoral factors but is also subject to a mechanical overload which is beneficial for bone health. Our finding that in prepubertal obese children with MetS bone formation still prevails over resorption is partly in line with some recent data showing that in adolescents, middle-aged adults and the elderly, metabolic syndrome is positively associated with bone mass or BMD [2,32,36].

Bone health may be assessed in different ways, and measurement of bone turnover markers, an easy to perform noninvasive procedure, is only one of the options. However, it has some advantages. The levels of serum bone turnover markers reflect the metabolism of the whole skeleton, and it has been suggested that changes in bone quality in some clinical conditions may be better explained using other bone outcome indicators than BMD [37]. A previous study indicated that in adolescents without MetS, bone turnover markers do not correlate with BMD, whereas in adolescent girls with MetS, a significant negative correlation was found [7]. For technical reasons, we were unable to perform BMD measurements, but we determined two bone turnover markers, i-P1NP and CTX-1, recommended as reference markers when assessing bone turnover in adults but nowadays frequently applied by several researchers investigating bone metabolism in children [11,12].

## 5. Conclusions

Metabolic complications that coexist with MetS may exert conjunctive effects on bone that may relate to subsequent bone development and are sex-and age-dependent. The present study suggests that dyslipidemia is not associated with levels of bone turnover markers in prepubertal children, and CRP, a marker of low-grade inflammation, is weakly related to CTX-1 only in girls. Our study also points out that sex difference is essential when the effect of insulin resistance is considered.

We suggest that in prepubertal girls, insulin resistance exerts a dominant impact on the decrease in bone resorption, independent of BMI centile and waist circumference. The bigger the worldwide obesity epidemic among children becomes, the more an in-depth explanation of complex effects of factors associated with metabolic syndrome on bone quality will require further investigation. Improvement of insulin sensitivity by dietary modifications and increasing physical activity may be of utmost significance for achieving adequate bone accrual in childhood and adolescence.

## Figures and Tables

**Figure 1 nutrients-14-01205-f001:**
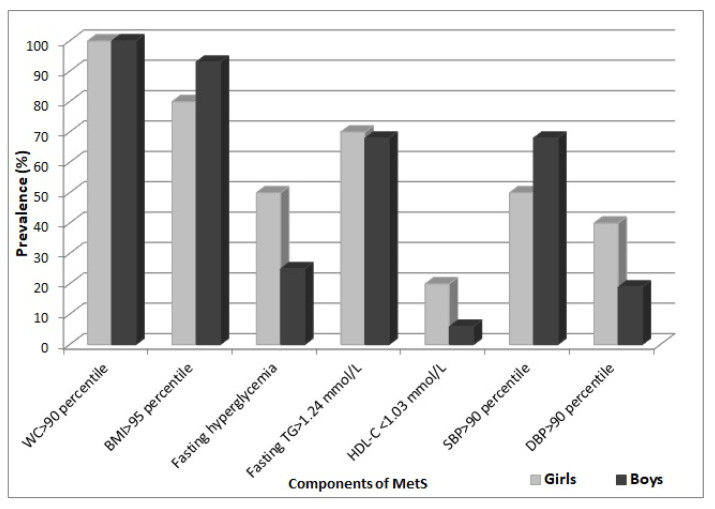
Percentage occurrence of components of MetS in girls and boys with MetS.

**Table 1 nutrients-14-01205-t001:** Clinical and biochemical characteristics of the study groups.

Variables	Girls without MetS *n* = 51	Girls with Central Obesity and ≥2 MetS Features *n* = 10	*p*	Boys without MetS *n* = 38	Boys with Central Obesity and ≥2 MetS Features *n* = 16	*p*
Age [years]	10 (9–10)	9 (9–10)	0.226	9.5 (9–10)	10 (9–10)	0.500
BMI centile	54 (19–64)	96 (95–97)	<0.001	36 (16–44)	96 (95–97)	<0.001
WC [cm]	63 (58–69)	80 (78–82)	<0.001	64 (60–67)	86 (83–89)	<0.001
Glucose [mmol/L]	5.1 (4.7–5.2)	5.5 (4.8–5.6)	0.098	5.0 (4.7–5.3)	5.1 (4.8–5.5)	0.205
Insulin [µIU/mL]	6.5 (4.2–10.0)	12.5 (11.3–13.7)	<0.001	9.1 (4.6–13.3)	17.6 (9.8–21.1)	<0.001
HOMA-IR	1.45 (0.93–2.30)	2.82 (2.50–3.47)	<0.001	1.93 (1.12–3.05)	4.51 (2.10–4.91)	<0.001
TG [mmol/L]	0.77 (0.54–0.95)	1.25 (1.17–1.39)	<0.001	0.62 (0.49–0.82)	1.37 (1.07–1.79)	<0.001
HDL-C [mmol/L]	1.53 (1.35–1.79)	1.35 (1.25–1.59)	0.048	1.56 (1.40–1.92)	1.27 (1.25–1.38)	0.009
SBP [mmHg]	107 (99–112)	116 (112–125)	0.001	108 (102–111)	116 (113–120)	0.004
DBP [mmHg]	63 (59–66)	72 (66–76)	<0.001	63 (60–67)	68 (65–72)	0.008
CRP [mg/L]	0.46 (0.18–1.21)	1.72 (0.75–1.86)	0.053	0.23 (0.14–1.10)	2.56 (1.21–3.03)	<0.001
i-P1NP [ng/mL]	1429 (1230–1581)	1383 (1141–1488)	0.291	1299 (1194–1449)	1252 (1208–1358)	0.506
CTX-1 [ng/mL]	1.65 (1.19–2.15)	0.98 (0.80–1.07)	0.001	1.59 (1.11–2.14)	1.24 (0.84–1.69)	0.057
BTI	0.26 (−0.41–0.85)	0.92 (0.61–1.55)	0.005	−0.28 (−0.95–0.37)	0.28 (−0.15–0.45)	0.103

Medians (25th and 75th percentiles) or percentage; WC—waist circumference; SBP—systolic blood pressure; DBP—diastolic blood pressure, TS—Tanner stage, BTI—bone turnover index, MetS—metabolic syndrome.

**Table 2 nutrients-14-01205-t002:** Percentage occurrence of increased values of biomarkers associated with MetS and decreased 25(OH)D in girls and boys with MetS.

Variable	Girls with MetS *n* (%)	Boys with MetS *n* (%)	*p*	*p* *
CRP > 3.0 mg/L	2 (20)	5 (31)	0.538	0.717
Fasting hyperinsulinemia	6 (60)	10 (63)	0.878	0.878
HOMA-IR > 3.0	5 (50)	10 (63)	0.513	0.717
25(OH)D < 20 ng/mL	2 (20)	5 (31)	0.538	0.717

MetS—metabolic syndrome, 25(OH)D—total 25-hydroxyvitamin D. *p* *—corrected *p*-value after applying Benjamini–Hochberg correction.

**Table 3 nutrients-14-01205-t003:** Associations of individual risk factors and biomarkers of bone metabolism with the presence of MetS in univariate logistic regression models stratified by sex.

Independent Variables	OR per 1SD Increase in Variable (95% CI) Girls, *n* = 61	*p*	*p* *	OR per 1SD Increase in Variable (95% CI) Boys, *n* = 54	*p*	*p* *
CRP	2.07 (1.03–4.14)	0.041	0.048	7.36 (2.2–24.2)	0.001	0.004
HOMA-IR	4.44 (1.70–11.61)	0.002	0.007	3.48 (1.50–8.08)	0.004	0.009
TG	8.23 (2.0–33.9)	0.004	0.007	17.6 (3.34–60.9)	<0.001	0.004
CTX-1	0.28 (0.11–0.69)	0.006	0.008	0.58 (0.31–1.062)	0.082	0.100
P1NP	0.71 (0.34–1.49)	0.371	0.371	0.83 (0.41–1.07)	0.613	0.613
SBP	6.53 (1.82–23.41)	0.004	0.007	2.71 (1.12–6.53)	0.026	0.036
DBP	4.06 (1.68–9.80)	0.002	0.007	2.50 (1.14–5.50)	0.023	0.036

SBP—systolic blood pressure; DBP—diastolic blood pressure; odd-ratio (OR); confidence interval (CI); standard deviation (SD); according to chi-squared statistic all models achieved statistical goodness of fit; *p* *—corrected *p*-value after applying Benjamini–Hochberg correction.

**Table 4 nutrients-14-01205-t004:** The multiple linear regression analysis between CTX-1 and HOMA-IR in models including BMI centile and WC in girls.

Independent Variables	R^2^	Beta (Standardized)	*p*	*p* *
Model 1	0.13		0.016	
HOMA-IR		−0.31	0.017	0.034
BMI centile		−0.11	0.385	0.385
Model 2	0.12		0.024	
HOMA-IR		−0.33	0.019	0.038
WC		−0.03	0.812	0.812

R^2^—R squared; *p* *—corrected *p*-value after applying Benjamini–Hochberg correction.

**Table 5 nutrients-14-01205-t005:** Associations of metabolic risk factors with CTX-1 values below the median (<1.50 ng/mL) in univariate logistic regression models stratified by sex.

Independent Variables	OR per 1SD Increase in Variable (95% CI) Girls *n* = 61	NR2	*p*	*p* *	OR per 1SD Increase in Variable (95% CI) Boys *n* = 54	NR2	*p*	*p* *
HOMA-IR	2.80 (1.42–5.52)	0.220	0.003	0.009	1.23 (0.74–2.06)	0.016	0.419	0.629
BMI centile	1.83 (0.98–3.42)	0.089	0.057	0.058	0.95 (0.53–1.70)	0.001	0.858	0.858
WC	1.67 (0.98–2.86)	0.083	0.058	0.058	1.16 (0.85–1.59)	0.007	0.348	0.629

WC—waist circumference odd-ratio (OR); confidence interval (CI); standard deviation (SD); NR2—Nagelkerke’s pseudo R^2^. *p* *—corrected *p*-value after applying Benjamini–Hochberg correction According to chi-squared statistics all models achieved statistical goodness of fit.

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
