# Peer review of "Relationships between Bone Turnover Markers and Factors Associated with Metabolic Syndrome in Prepubertal Girls and Boys"

_nutrients, 2022, doi:10.3390/nu14061205_

Round 1
Reviewer 1 Report
The present study investigates the association of factors associated with metabolic syndrome and bone turnover markers in children, highlighting sex differences. The subject of this study is interesting due to the contradictory results of previous studies. However, there are some points I would like to share with the authors:
Major points
- Was the study registered?
- Please add some lines in the introduction about parameters used to evaluate bone turnover (P1PN and CTX).
- In the first part of the results section, results from the descriptive analysis are considered. It would be more interesting and correct to show the results of adjusted models. Moreover, physical activity and dietary information should be included in the models since these factors influence metabolic syndrome and are of “utmost significance for achieving an adequate bone accrual in childhood and adolescence”. If this data is not available, please add a point among the limits of the study.
- Statistical tests for descriptive tables are not necessary. The sample size is small in the different sub-groups so it will be hard to reject the null hypothesis even in the case of conspicuous differences.
- Since you considered different individual risk factors, in the statistical analysis, have you considered correction for multiple comparisons?
Minor points
- Please verify that the abbreviations are used after the first time they appear in the text (i.e., line 146 CTX).
- Did the authors verify normal distribution after log transformation?
- Line 158. Referred to Figure 1, please specify that “Among the components of MetS in both girls and boys, fasting hypertriglyceridemia was the most prevalent, followed…” apart of central adiposity.
- Results of univariate analysis are not shown.
- Why data of adjusted models are not shown?
Author Response
We submit the revised version of the manuscript
Relationships between bone turnover markers and factors associated with metabolic syndrome in prepubertal girls and boys.
by Wojciech J. Bilinski*, Anna Stefanska*, Lukasz Szternel, Katarzyna Bergmann, Joanna Siodmiak, Magdalena Krintus, Przemyslaw T. Paradowski, and Grazyna Sypniewska
We want to extend our appreciation to the reviewers and editorial board for taking the time and effort necessary to improve our work and provide such insightful guidelines.
We have carefully considered all comments. Please find attached the detailed reply to all comments/suggestions.
Our responses to the comments of the reviewer 1 are presented in italics.
Reviewer 1
Major points
- Was the study registered?
No, this was a small research study of a purely observational design which according to our University regulations does not require registration. Our study was conducted in accordance with the Declaration of Helsinki and our previous cross-sectional was approved by the Collegium Medicum Bioethics Committee at the Nicolaus Copernicus University (KB 348/21.04.2015). All participants and their parents were thoroughly informed about all aspects of the study, and written informed consent was obtained.
- Please add some lines in the introduction about parameters used to evaluate bone turnover (P1PN and CTX).
In the Introduction we have added the most relevant information on both parameters (P1NP and CTX) used to evaluate bone turnover in children, lines 74-95.
- In the first part of the results section, results from the descriptive analysis are considered. It would be more interesting and correct to show the results of adjusted models. Moreover, physical activity and dietary information should be included in the models since these factors influence metabolic syndrome and are of “utmost significance for achieving an adequate bone accrual in childhood and adolescence”. If this data is not available, please add a point among the limits of the study.
Unfortunately, we were unable to collect detailed dietary information and information on the physical activity outside the school. The limitations of the study were extended according to reviewer’s suggestion (lines 361-364).
- Statistical tests for descriptive tables are not necessary. The sample size is small in the different sub-groups so it will be hard to reject the null hypothesis even in the case of conspicuous differences.
The study group is indeed small - this information was added to the limits of the study (“the small sample size, in particular very small numbers of girls and boys with metabolic syndrome which makes the problem in rejecting the null hypothesis in comparison of subgroups and drawing general conclusions quite difficult”).
However, it is worth noting that the occurrence of MetS in Polish children is estimated at approx. 10-15%, so in the group of healthy children that we recruited for the previous study, individuals with MetS components constituted a small part of the participants. Although statistical analysis on small groups has limitations (the problem in rejecting the null hypothesis), we decided to compare the groups using nonparametric tests to show the significance of the differences. On the other hand, due to small sample size, we did not use the adjusted models in the comparative and logistic regression analysis. We only decided to use two independent variables in multiple regression models to evaluate that the correlation between CTX-1 and HOMA-IR persists after controlling for BMI centile or WC in girls.
- Since you considered different individual risk factors, in the statistical analysis, have you considered correction for multiple comparisons?
To correct p values the Benjamini-Hochberg procedure was applied to risk factor results aiming to reduce the potential for Type 1 error.
Minor points
- Please verify that the abbreviations are used after the first time they appear in the text (i.e., line 146 CTX).
We have verified the abbreviations throughout the manuscript, particularly for the bone markers. In the text we used the abbreviations i-P1NP and CTX-1 when referring to the results of our study which is due to the names given by the manufacturer of the tests; we have measured intactP1NP (i-P1NP) and CTX-1. Referring to the literature data we used the abbreviations P1NP and CTX/CTx.
- Did the authors verify normal distribution after log transformation?
Yes, we verified. All parameters achieved the normal distribution after log transformation.
- Line 158. Referred to Figure 1, please specify that “Among the components of MetS in both girls and boys, fasting hypertriglyceridemia was the most prevalent, followed…” apart of central adiposity.
Thank you very much for this important comment. The sentence was corrected accordingly (line 206).
- Results of univariate analysis are not shown.
The results of univariate analysis are shown in table 3 and 5.
- Why data of adjusted models are not shown?
The results of adjusted models are shown in table 4.
Reviewer 2 Report
In this study, authors evaluated some of the bone turnover markers, and assessed connections with metabolic syndrome in prepubertal children.
Study is interesting, with some valuable results, however several aspects should be further addressed before potential publication:
Introduction section could use a better flow of information, as there are no paragraphs, and studies that are listed and commented on appear without observable order. Also, it is necessary to further emphasize clear gap in the literature and novelty of this study – in this introduction structure, it is not clear enough.
Even though this sample is a cohort that is enrolled in a previous study, main protocol and description of children inclusion in this study should be mentioned, as well as detailed sample size calculation.
More detailed rationale why authors chose exactly P1NP marker of bone formation, and not e.g. OC or B-ALP needed. It would be beneficial for the study that all of the markers were assessed, as they have pathophysiological differences
P value should always be written in 3 decimal places, in the Results
Table 1 is large and not reader-friendly – it should be amended, or split into more different tables in a logical manner
Figure 1 caption should be put outside the figure.
Add paragraphs in Discussion section, it is difficult to read without them – like in the introduction, flow of information is obstructed and it is somewhat difficult to read the text
Authors could put more emphasis into offering possible explanations and hypotheses of the exact results of bone turnover parameters associations with other variables
In Discussion, more emphasis should be put into possible clinical implications of the results of this study
In general, bone turnover markers have high biovariability - this should be better explained as a possible limitation of the study
Author Response
We submit the revised version of the manuscript
Relationships between bone turnover markers and factors associated with metabolic syndrome in prepubertal girls and boys.
by Wojciech J. Bilinski*, Anna Stefanska*, Lukasz Szternel, Katarzyna Bergmann, Joanna Siodmiak, Magdalena Krintus, Przemyslaw T. Paradowski, and Grazyna Sypniewska
We want to extend our appreciation to the reviewers and editorial board for taking the time and effort necessary to improve our work and provide such insightful guidelines.
We have carefully considered all comments. Please find attached the detailed reply to all comments/suggestions.
Our responses to the comments of the reviewer 2 are presented in italics.
Reviewer 2
Introduction section could use a better flow of information, as there are no paragraphs, and studies that are listed and commented on appear without observable order. Also, it is necessary to further emphasize clear gap in the literature and novelty of this study – in this introduction structure, it is not clear enough.
We agree with the reviewer’s comments and the Introduction has been rewritten to ensure better flow of information. We also tried to emphasize the novelty of our research (lines 70-71 and 91-95). We believe the new structure of Introduction fulfills reviewer’s expectation
Even though this sample is a cohort that is enrolled in a previous study, main protocol and description of children inclusion in this study should be mentioned, as well as detailed sample size calculation.
According to the reviewer’s suggestion we have included the protocol of the previous study and presented the inclusion criteria for the study participants (lines 101-111). From a cohort of 284 presumably healthy school children aged 9-11, enrolled in a previous cross-sectional study, a subset of 115 children was selected for whom complete data on biochemical parameters, markers of bone metabolism, blood pressure and anthropometric parameters were available. In this subset of 115 children : 89 subjects (51 girls and 38 boys) presented without any feature of metabolic syndrome (MetS) and 26 subjects (10 girls and 16 boys) presented with central obesity and at least two other features of MetS. We believe our explanation on the sample size is satisfying.
More detailed rationale why authors chose exactly P1NP marker of bone formation, and not e.g. OC or B-ALP needed. It would be beneficial for the study that all of the markers were assessed, as they have pathophysiological differences.
We agree it could be beneficial to study also other markers engaged in bone formation however, many recent studies conducted in children and adolescents applied P1NP to investigate bone metabolism. We have chosen P1NP as the marker of bone formation as this marker is recommended as a reference marker by Int Osteoporosis Foundation and IFCC. P1NP is the only bone formation marker that reflects the synthesis of new bone collagen and its serum level displays greater dynamic changes than other bone formation markers in response to changes of bone metabolism. P1NP molecule is well characterized, has a very little diurnal and biological variation. Osteocalcin shows similar biological variation but has a circadian rythm. B-ALP has a little diurnal variation but its level has been shown affected by several life style factors that do not influence P1NP. The levels of P1NP are very stable between age 4-13th (present very narrow range of changes) whereas OC starts to increase after 11 years of age in girls and after 13 years of age in boys and its concentrations change within a wide range from 4 till 14 years and 16 years in females and males, respectively. This has been addressed by us in the manuscript (lines 295-303).
P value should always be written in 3 decimal places, in the Results
In the Results section all p values were corrected and presented with 3 decimal places.
Table 1 is large and not reader-friendly – it should be amended, or split into more different tables in a logical manner
Table 1 has been rearranged and we think now is reader-friendly.
Figure 1 caption should be put outside the figure.
Fig 1 has been changed, accordingly.
Add paragraphs in Discussion section, it is difficult to read without them – like in the introduction, flow of information is obstructed and it is somewhat difficult to read the text
We agree with the reviewer’s suggestions and the Discussion was restructured with paragraphs added.
Authors could put more emphasis into offering possible explanations and hypotheses of the exact results of bone turnover parameters associations with other variables.
We thank for this comment and assume it refers to the possible associations of bone turnover with blood pressure which was shown as one of the predictors of having MetS in our children. The possible explanation is not easy to present as we have not observed meaningful correlations between SBP or DBP and bone markers in our small subgroups of children with MetS. We have included our comments referring to the possible effect of blood pressure on bone health on the basis of data from adults and conflicting data from adolescents (lines344-357).
In Discussion, more emphasis should be put into possible clinical implications of the results of this study
We believe that the last sequence of our conclusions fulfill reviewer’s expectations (lines 397-402).
In general, bone turnover markers have high biovariability - this should be better explained as a possible limitation of the study
According to data by Cavalier et al. European Biological Variation Study stablished for healthy adults (Osteoporos. Int 2020, ref. 27) within subject biological variation for P1NP and CTX are [8.8 (8.4-9.3%CV) and 15.1(14.4-16.0%CV), respectively whereas reference change values (%) are higher (up to 24.8% for P1NP and up to 44.5% for CTX) however, as we mentioned above the levels of both bone markers we assessed are very stable for P1NP between 9-11 years and quite stable for CTX. We explained this in the Discussion (lines 364-366) and deeply believe this is a strenght of our study.
Round 2
Reviewer 2 Report
The authors have been highly responsive, and have amended the manuscript according to the comments.
I have no further questions.